# Polystyrene microplastic particles induce endothelial activation

**Ann-Kathrin Vlacil**[1], **Sebastian Bänfer**[2], **Ralf Jacob**[2], **Nicole Trippel**[1], **Istemi Kuzu**[3], **Bernhard Schieffer**[1], **Karsten Grote**[1]*

1 Cardiology and Angiology, Philipps-University Marburg, Marburg, Germany, 2 Department of Cell Biology and Cell Pathology, Philipps-University Marburg, Marburg, Germany, 3 Department of Chemistry, Philipps-University Marburg, Marburg, Germany

* karsten.grote@staff.uni-marburg.de

**Data Availability Statement:** All relevant data are within the paper and its Supporting Information files.

**Funding:** BS, KG—Dr. Reinfried Pohl-Stiftung https://www.dr-reinfried-pohl-stiftung.de/ The funders had no role in study design, data collection

## Abstract

Due to its increasing production, durability and multiple applications, plastic is a material we encounter every day. Small plastic particles from the µm to the mm range are classified as microplastics and produced for cosmetic and medical products, but are also a result of natural erosion and decomposition of macroplastics. Although being omnipresent in our environment and already detected in various organisms, less is known about the effects of microplastics on humans in general, or on vascular biology in particular. Here we investigated the effects of carboxylated polystyrene microplastic particles (PS, 1 µm) on murine endothelial and immune cells, which are both crucially involved in vascular inflammation, using *in vitro* and *in vivo* approaches. *In vitro*, PS induced adhesion molecule expression in endothelial cells with subsequent adhesion of leukocytes both under static and flow conditions. In monocytic cells, PS enhanced pro-inflammatory cytokine expression and release. Accordingly, administering mice with PS led to enhanced aortic expression of cytokines and adhesion molecules. Furthermore, we identified neutrophils as the PS-clearing blood leukocyte population. The findings from this study for the first time indicate polystyrene microplastic as a new environmental risk factor for endothelial inflammation.

## Introduction

Since the last decades, living beings are increasingly faced with several novel environmental risk factors, such as particulate matter, plasticizers and microplastics. The latter are small synthetic and solid particles with a size ranging from less than 1 µm to 5 mm and can be divided in primary microplastics produced for instance for cosmetic and medical products and in secondary microplastics resulting from natural erosion and decomposition of macroplastics [1]. An example for the latter is polystyrene (PS) being present in particulate matter because of tire abrasion. Microplastics continuously accumulate in our environment due to the globally growing production of plastic reaching 368 million tons in 2019 according to the association of plastics manufacturers [2]. There is no doubt that plastic is a material of great benefit at relatively low production cost and offers a wide range of applications. Hence, the collective term

and analysis, decision to publish, or preparation of the manuscript.

**Competing interests:** The authors have declared that no competing interests exist.

plastic includes various polymers with polypropylene, polyethylene and polystyrene being the most widely used representatives in everyday life in regard to food packaging, disposable plastic or tire abrasion [2]. Microplastics have been detected in coastal regions, oceans and also in marine organisms like phytoplankton, mussels and fish, thus ultimately entering the human food chain [3–5]. Accordingly, a study tested human stool samples of different regions around the world positive for major microplastic polymers such as PS particles [6]. Of note, just recently the same group detected microplastic particles in human placenta and meconium as well [7]. Therefore, microplastics represent a novel environmental factor; however, their effects on human health are largely unknown. To date, only few *in vivo* studies investigated microplastics in this regard. Recent studies reported microplastic particle accumulation in different organs like liver and heart upon oral administration [8–10], demonstrating their uptake via the intestine into the circulation where they may interact with immune cells and the endothelium promoting inflammatory effects [11]. Additionally, PS particles have been shown to impair lipid metabolism in murine macrophages [12] and to trigger hepatoxicity [13]. In the context of a potential use of PS particles as drug carriers, Barshtein et al. observed elevated aggregation and endothelial adhesion of red blood cells treated with PS *in vitro* [14]. However, effects on the vasculature are largely unknown. For the first time, we show the capability of carboxylated PS particles to activate endothelial cells and enhance monocyte adhesion. In addition, we demonstrate their uptake by neutrophils in the blood.

## Materials and methods

### Cell culture and treatment

For *in vitro* experiments, we used murine cell lines: endothelial MyEND cells (myocardial endothelial cells, RRID: CVCL_2131) which we recently characterized in regard to endothelial markers and properties [15] as well as monocytic J774A.1 cells (CVCL_0358), which were cultured in a humidified incubator at 37˚C and 5% $CO_2$. MyEND cells were cultured using Dulbecco´s Modified Eagle´s Medium (DMEM, Gibco, Darmstadt, Germany) supplemented with 10% fetal calf serum (FCS, PAN-Biotech, Aidenbach, Germany) and 1% Penicillin/Streptomycin (PenStrep, 100 U/mL and 100 mg/mL, Sigma-Aldrich, Seelze, Germany). For J774A.1, DMEM GlutaMAX (Gibco) supplemented with 10% FCS and 1% PenStrep was used. PS particles conjugated with tetramethylrhodamine isothiocyanate (TRITC) or unconjugated (1 µm, carboxylated, Kisker Biotech, Steinfurt, Germany,) were used as a representative of microplastics abundantly occurring in our environment. Polystyrene as polymer component of the particles was confirmed by Raman spectroscopy (S1A and S1B Fig) and size distribution of the PS particles were measured by dynamic light scattering (1108±185.6 nm, Zetasizer Nano ZS90, Malvern Pananalytical, Malvern, UK) (S1C Fig). Using the above mentioned medium supplemented with 1% FCS, J774A.1 (sub-confluent) and MyEND (confluent) cells were starved for 2 hours and subsequently stimulated with PS particles ($10^3$, $10^5$ and $10^7$ particles/mL, which corresponds to 0.54 ng/mL, 54 ng/mL and 5.4 µg/mL) for 3 and 6 hours in a 12-well plate, respectively. RNA and supernatant were used for further real-time PCR and ELISA analysis.

### Cell viability assay

Cell viability of J774A.1 and MyEND cells after 16 hours of exposure to PS particles was analyzed using the alamarBlue Cell Viability Assay (Thermo Fisher Scientific, Waltham, MA, USA) according to manufacturer´s instructions. Both cell types were exposed to an increasing concentration of PS. Triton was used as positive control.

## Animal experiments

Wild type C57BL/6N mice at the age of 10–12 weeks were given 2.5 mg fluorescent PS-TRITC particles (1 μm, carboxylated, Kisker Biotech) or PBS (up to 9 mice per group) as control by intravenous injection. All animal experiments were approved by the regional council Giessen (99/2019) and conform to the guidelines from directive 2010/63/EU of the European Parliament. After 3 hours, mice were anesthetized using Ketamin and Xylazin (120 mg and 12 mg/kg bodyweight) and euthanized by puncturing the left ventricle. Peripheral blood, liver and aortic tissue (aortic arch and thoracoabdominal aorta) were collected. Blood and liver samples were used for immunohistochemistry, aortic tissue for gene expression analysis.

## Real-time PCR

For the analysis of mRNA expression, total RNA from aortic and liver tissue, J774A.1 and MyEND cells was isolated using RNA-Solv® Reagent (Omega Bio-tek, Norcross, GA, USA) following manufacturer´s instructions and reverse-transcribed with SuperScript reverse transcriptase, oligo(dT) primers (Thermo Fisher Scientific) and deoxynucleoside triphosphates (Promega, Madison, WI, USA). Real-time PCR was performed in duplicate in a total volume of 20 μL using Power SYBR Green PCR master mixture (Thermo Fisher Scientific) on a Step One Plus real-time PCR system in 96-well PCR plates (Applied Biosystems, Waltham, MA, USA). SYBR Green emissions were monitored after each cycle. For normalization, expression of *Gapdh* was determined in duplicates. Relative gene expression was calculated by using the $2^{-\Delta\Delta Ct}$ and $2^{-\Delta Ct}$ method for matched and independent pairs sampling respectively. Real-time PCR primers for mouse *Il-1β*, *Tnf-α*, *Vcam-1*, *Icam-1*, *Saa1*, *Saa2*, *Saa3* and *Gapdh* were obtained from Microsynth AG (Balgach, Switzerland): *Il-1β*-F: 5′-GCC ACC TTT TGA CAG TGA TGA G-3′, *Il-1β*-R: 5′-GAC AGC CCA GGT CAA AGG TT-3′, *Tnf-α*-F: 5′-CTG GCA CCA CTA GTT GGT TGT-3′, *Tnf-α*-R: 5′-GTA GCC CAC GTC GTA GCA AAC-3′, *Vcam-1*-F: 5′-TCT TAC CTG TGC GCT GTG AC-3′, *Vcam-1*-R: 5′-ACC TAG CGA GGC AAA CAA GA-3′, *Icam-1*-F: 5′-CAC GTG CTG TAT GGT CCT CG-3′, *Icam-1*-R: 5′-TAG GAG ATG GGT TCC CCC AG-3′, *Saa1*-F: 5′-CCC AGG AGA CAC CAG GAT GA-3′, *Saa1*-R: 5′-TCA TGT CAG TGT AGG CTC GC-3′, *Saa2*-F: 5′-TGC TGA GAA AAT CAG TGA TGC AA-3′, *Saa2*-R: 5′-CCC AAC ACA GCC TTC TGA AC-3′, *Saa3*-F: 5′-GAA AGA AGC TGG TCA AGG GTC-3′, *Saa3*-R: 5′-TCC GGG CAG CAT CAT AGT TC-3′, *Gapdh*-F: 5′-GTC TCC TGC GAC TTC AGC-3′, *Gapdh*-R: 5′-TCA TTG TCA TAC CAG GAA ATG AGC-3′.

## Enzyme-linked immunosorbent assay (ELISA)

Supernatant of J774A.1 and MyEND cells were analyzed for TNF-α and sVCAM-1 respectively, using mouse-specific ELISA (R&D Systems, Minneapolis, MN, USA) according to manufacturer's protocol with the help of an Infinite M200 PRO plate reader (TECAN Instruments, Waltham, MA, USA).

## Adhesion assay

**Adhesion under static conditions.**  MyEND cells were grown to complete confluence in a 24-well plate. Cells were stimulated in DMEM supplemented with 1% FCS for 16 hours with PS ($10^7$ particles/mL). Untreated cells cultured in DMEM supplemented with 1% FCS served as control. J774A.1 cells were labelled with 1 μM of Calcein AM (eBioscience, San Diego, CA, USA, excitation: 495 nm, emission: 515 nm). After stimulation, MyEND cells were washed twice with 500 μL DMEM and $0.5 \times 10^6$ labelled J774A.1 cells were added per well and

incubated for 1 hour in 5% $CO_2$ at 37°C. After incubation each well was washed three times with 500 µL DMEM and 10 high power field digital images were taken using Axio Vert.A1 microscope with an AxioCam MRm camera (Carl Zeiss, Oberkochen, Germany). Adhered cells per HPF were counted using ImageJ software (National Institute of Health, Bethesda, MD, USA).

**Adhesion under flow conditions.** For flow based adhesion assay, $5x10^5$ MyEND cells were plated in µ-Slide VI$^{0.4}$flow chamber (ibidi, Martinsried, Germany) in DMEM supplemented with 10% FCS and 1% PenStrep and grown to complete confluence. Cells were stimulated as described above. After stimulation, µ-Slide was connected to a 20 cm silicon tubing attached to a luer lock adaptor, which was in turn connected to a 50 mL syringe attached to a Perfusor VII pump (B Braun AG, Melsungen, Germany). For flow conditions, a flow rate of 0.53 mL/min (which corresponds to laminar flow of 0.5 dyne/cm$^2$) was maintained. Endothelial layer was perfused with DMEM for 2 min to remove any debris and dead cells. After washing, Calcein-labeled J774A.1 cells at a concentration of $1x10^6$ cells/mL were perfused for 2 hours at a constant shear stress of 0.5 dyne/cm$^2$, followed by perfusion with DMEM for 30 min to remove unbound cells. The last 2 min of washing were recorded by a DM550B fluorescence microscope with a DFC300FX camera (Leica Microsystems, Wetzlar, Germany) and 20 HPF digital images from this recording were subsequently used for the analysis. Adhered cells per high power field were counted using ImageJ software.

## Immunohistochemistry and imaging

Liver tissue and peripheral blood from wild type mice administered with PS-TRITC or PBS was collected and erythrocytes were lysed. Blood smear samples and liver cryosections were prepared and PS-TRITC particles were subsequently observed using an Axio Vert.A1 microscope with an AxioCam MRm camera (Carl Zeiss). Blood smear samples were stained for DAPI and rat anti-mouse Ly6G (Invitrogen, Waltham, MA, USA). Goat anti-rat IgG conjugated to FITC (Invitrogen) was used as secondary antibody. Liver sections were stained for DAPI and Alexa Fluor 488-WGA (Thermo Fisher Scientific). Confocal images were acquired on a Leica TCS SP2 microscope. Image analysis was performed using Leica LAS AF and ImageJ.

## Statistical analysis

All data are represented as means ± SD. Groups were compared using parametric 2-tailed Student t-test (GraphPad Prism, version 9.01; GraphPad Software, La Jolla, CA, USA). A value of $P < 0.05$ was considered statistically significant. Real-time PCR was performed in technical duplicates.

## Results

In initially performed studies assessing cell viability using both murine monocyte/macrophage J774A.1 cells and endothelial MyEND cells, PS particles with a concentration ranging from $10^3–10^7$ particles/mL did not cause any cytotoxic effects in these cells after 16 hours, the longest period of stimulation with PS particles in this study (Fig 1A). We performed pilot studies in order to define time- and dose-dependence for both cell lines and observed greatest effects on inflammatory cytokine expression with a stimulation period of 3 hours for J774A.1 and 6 hours for MyEND cells, respectively using $10^7$ particles/mL (S2A Fig). At this concentration, a monolayer of MyEND cells is sparsely covered with single, doublets or small aggregates of PS particles ensuring appropriate particle/cell contact under our experimental conditions (S2B Fig). In J774A.1 cells, PS particles induced the expression of inflammatory cytokines such as *Il1-β* and *Tnf-α* as well as TNF-α release. In endothelial MyEND cells, PS particles up-

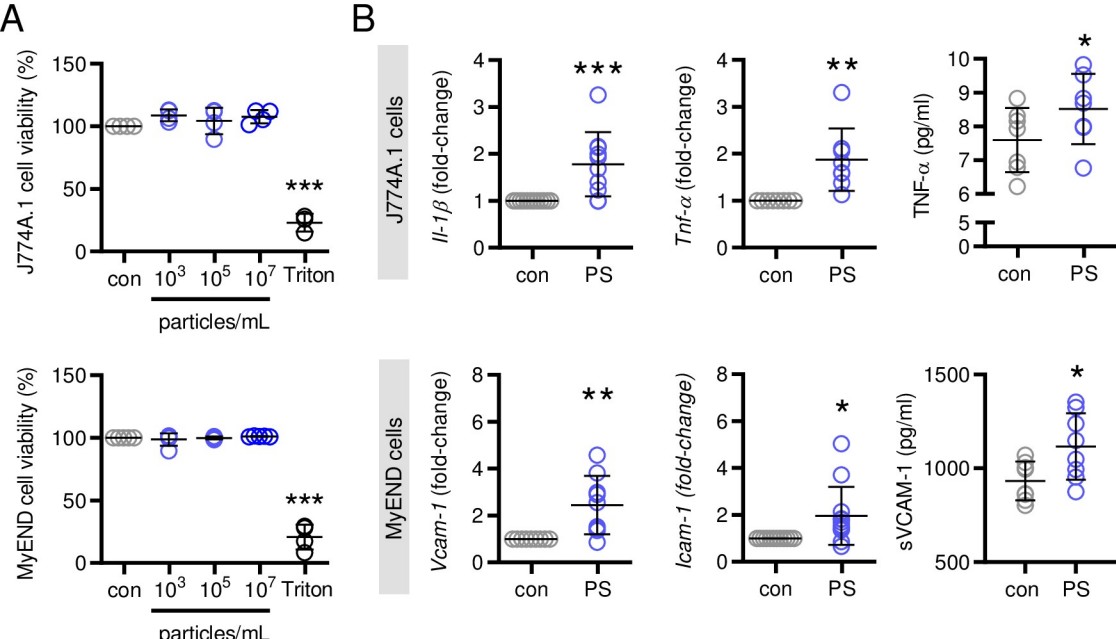

**Fig 1. Effects of PS particles on cell viability and inflammatory gene expression.** (**A**) Cell viability (%) of J774A.1 and MyEND cells after 16 hours of PS particles stimulation ($10^3$–$10^7$ particles/mL) was determined using alamarBlue cell viability assay, n = 3–5. (**B**) *Il1-β*, *Tnf-α* expression and TNF-α levels after 3 hours of PS particles stimulation ($10^7$ particles/mL) in J774A.1 cells (above) and *Vcam-1*, *Icam-1* expression and sVCAM-1 levels after 6 hours of PS stimulation ($10^7$ particles/mL) in MyEND cells (below) were determined by real-time PCR and by ELISA, respectively, n = 7–10. *P<0.05, **P<0.01, ***P<0.001 vs. control (con). Data were analyzed by Student t-test and are depicted as mean±SD.

regulated the expression of the major endothelial adhesion molecules vascular cell adhesion molecule (*Vcam*)-1 and intercellular adhesion molecule (*Icam*)-1 indicative of endothelial activation. Moreover, stimulation with PS particles increased sVCAM-1 release from endothelial cells (Fig 1B).

To investigate a potential functional effect of PS particles on monocyte adhesion under static and flow (0.53 mL/min) conditions, we incubated confluent MyEND cells with PS particles for 16 hours to allow adequate expression of the adhesion molecules at the protein level on the cell surface. Subsequently, we added Calcein-AM-labelled J774A.1 cells, followed by a standardized washing protocol. Following PS particle stimulation, significantly more adherent monocytic J774A.1 cells were detected under static as well as under flow conditions by fluorescence microscopy (Fig 2A and 2B).

For *in vivo* studies, C57BL/6N wild type mice were administered with 4.65x$10^9$ TRITC-conjugated PS particles (corresponds to 2.5 mg, 1 μm, Kisker Biotech). PS particles were consequently detected in the liver by confocal microscopy (Fig 3A) and significantly up-regulated the hepatic expression of serumamyloid A (*Saa*)1, *Saa*2 and *Saa*3 (Fig 3B), indicating an inflammatory acute-phase response of the liver to PS particles [16].

To characterize potential interactions of PS particles with circulating leukocytes, blood smear samples were prepared after 3 hours of administration. Free PS particles as well as accumulated PS particles taken up by Ly6G-positive cells were detected in the peripheral blood, identifying neutrophils as a particles-clearing blood leukocyte population (Fig 4A). In line with our findings *in vitro*, we determined significantly enhanced levels of the cytokine *Il-1β* as well as the endothelial activation markers *Vcam-1* and a trend (P = 0.069) towards elevated *Icam-1* expression in the aortic tissue from mice injected with PS particles (Fig 4B).

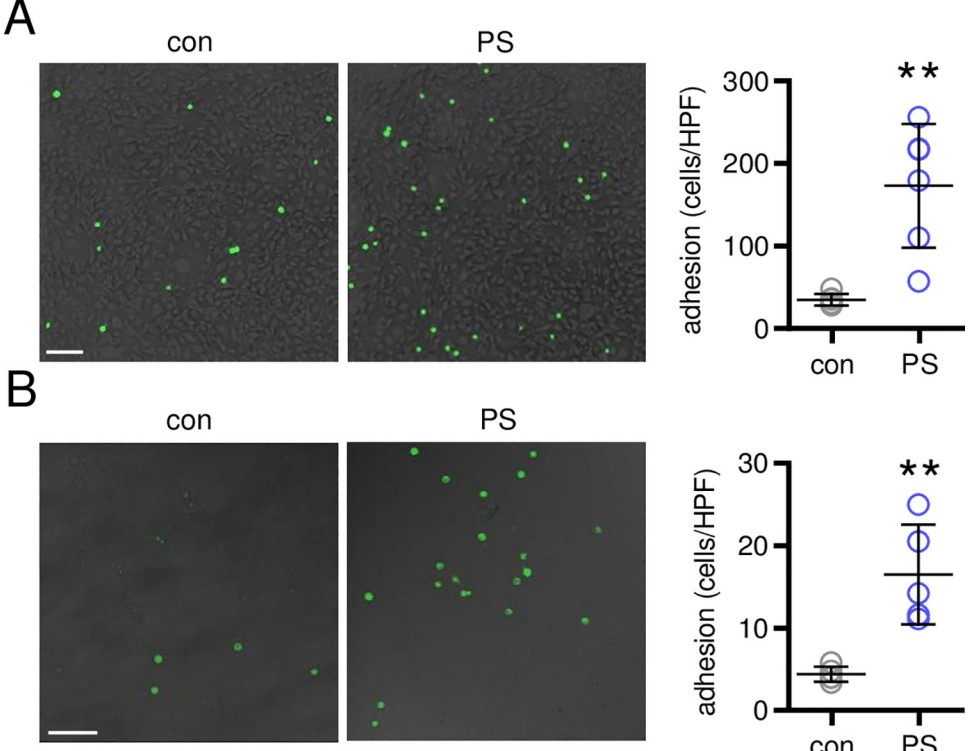

**Fig 2. Effects of PS particles on monocytic cell adhesion to endothelial cells.** Fluorescence microscopy images depicting Calcein-AM-labelled J774A.1 cells on a MyEND monolayer after stimulation with PS particles for 16 hours ($10^7$ particles/mL), one hour adhesion under static (**A**) and two hours under flow (**B**) conditions and subsequent washing. Scale bars = 100 μm. Adherent cells per high power field (HPF) were quantified. N = 4–6. **P<0.01 vs. control (con). Data were analyzed by Student t-test and are depicted as mean±SD.

## Discussion

Microplastic pollution is an increasing global challenge that raises concerns about our health as well. First studies show the translocation of these particles across the intestinal barrier into

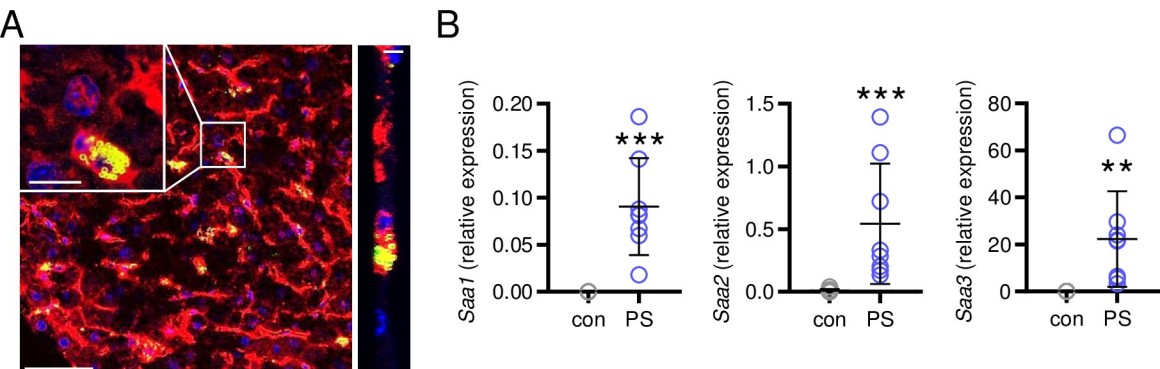

**Fig 3. Effects of PS particles *in vivo*: Hepatic accumulation and mRNA expression.** (**A**) Confocal microscopy composite of liver sections from mice 3 hours after injection with 2.5 mg TRITC-conjugated PS particles. blue: DAPI, red: Alexa Fluor 488-WGA, green: TRITC-PS (appear intracellularly in yellow). Scale bars: overview = 50 μm, insertion = 10 μm, z-stack in right panel = 5 μm. (**B**) Hepatic *Saa1*, *Saa2* and *Saa3* expression of mice administered with either 2.5 mg TRITC-conjugated PS particles or PBS as control were determined by real-time PCR, n = 8–9 mice/group. **P<0.01, ***P<0.001 vs. control (con). Data were analyzed by Student t-test and are depicted as mean±SD.

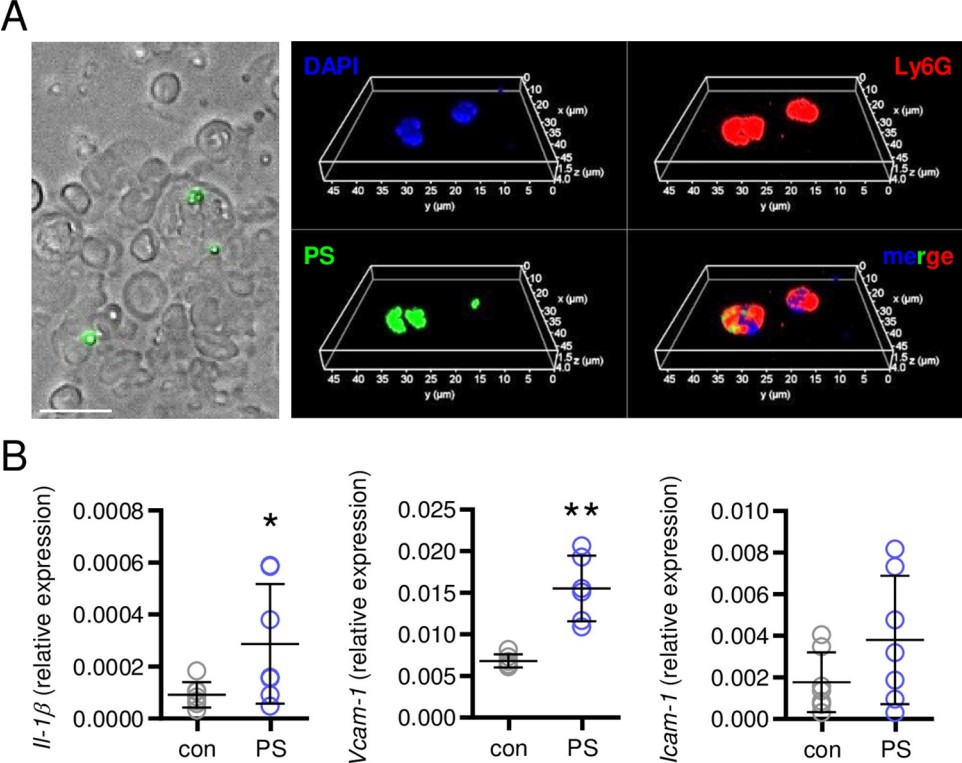

**Fig 4. Effects of PS particles *in vivo*: Peripheral blood and aortic tissue. (A)** Overlay of brightfield and epiflourescence (left) and rendered confocal image stack (right) of peripheral blood cells 3 hours after administration of 2.5 mg TRITC-conjugated PS particles. Whole blood was collected and red blood cells were lysed prior to staining. Scale bar = 10 μm. Blue: DAPI, red: FITC-Ly6G, green: TRITC-PS. **(B)** *Il-1β*, *Vcam-1* and *Icam-1* expression in aortic tissue of mice administered with either 2.5 mg TRITC-conjugated PS particles or PBS as control were determined by real-time PCR, n = 7 mice/group. *P<0.05, **P<0.01 vs. control (con). Data were analyzed by Student t-test and are depicted as mean±SD.

the organism. What consequences on biological processes this uptake may have and whether the particles thus initiate pathological processes is largely unknown. Only few studies investigated the effects of mainly polystyrene so far, for instance showing cytotoxic effects on the gut barrier function [17] and lung epithelial cells [18] *in vitro*. Performing a long-term exposure of the marine organism *daphnia magna* to polystyrene microplastic, Trotter et al. recently showed an impact on morphology, number of offspring and the organisms´ proteome, suggesting an altered supply for nutrients and overall fitness [19]. Using the same model organism, Kelpsiene et al. reported a reduced life-time upon exposure to polystyrene nanoplastics [20]. Of note, some studies reported translocation in a surface charge-dependent manner in both marine organisms and rats [8, 21]. Based on these observations and the fact that due to weathering processes and subsequent surface oxidation, microplastic particles are expected to acquire carboxylic groups (-COOH) and hence are negatively charged [22], we therefore have used carboxylated PS particles in our study as well.

We are aware that our study has some limitations, e.g. regarding dosage and route of exposure. Until today, there are only speculations about the actual exposure and uptake of microplastics into the human body, ranging from ng to mg per month. Accordingly, using 2.5 mg of particles for *in vivo* studies simulates an extreme exposure and certainly does not adequately reflect long-term, low-dosage microplastics intake to which most people in western countries are exposed. Additionally, the vast amount of microplastic particles in the environment is

considered to be taken up by inhalation or ingestion. Thus, using an intravenous injection for administration as in our study indicates an artificial exposure. Yet, we decided for this experimental setup to provide first data on the effects of PS particles being present in the circulation, thereby directly interacting with leukocytes and endothelial cells. In addition, we are only using one type of well-defined microplastic particles (PS, 1 μm, carboxylated), whereas microplastics in our environment are very heterogeneous in terms of material, size, surface texture and charge. In the environment, microplastic particles absorb many other potentially toxic substances such as nitrogen oxides, bacterial products, toxic organic chemicals or heavy metals due to their surface properties [22]. Thus, their inflammatory effects are probably higher than that of the sterile particles used in this study. Further studies are needed to evaluate the effects of low-dosage and long-term administration of microplastic particles of different material using for instance an oral administration route in preclinical models, also in regard to vascular inflammation and disease.

Our data show enhanced inflammatory cytokine and adhesion molecule expression *in vitro* and *in vivo*. These findings are in line with other *in vitro* studies observing enhanced intestinal gene transcription of classical inflammatory pathways like nuclear factor kappa B (NF-κB) and mitogen-activated protein kinase (MAPK) [17] or production of reactive oxygen species and elevated lipotoxicity in macrophages [12] after stimulation with polystyrene. Furthermore, *in vivo* we observed an increase in hepatic mRNA expression of the acute-phase proteins *Saa1*, *Saa2* and *Saa3* after polystyrene microplastic administration, representing a well-known reaction of the liver to inflammatory stimuli [16]. This observation is consistent with other studies showing hepatoxic effects of orally administered polystyrene in mice and rats [8, 23]. Of note, it was shown that photodegradation of polystyrene microplastic results in elevated hepatic inflammation and toxicity in the marine organism *Epinephelus moara*, suggesting that weather conditions such as solar radiation could even increase the hazard potential of polystyrene microplastic on aquatic organisms [24, 25].

In addition to already identified triggers of vascular inflammation including bacterial components during infection, reactive oxygen species or modified lipoproteins, this study suggests PS particles to be another, yet not described trigger initiating endothelial cytokine expression and leukocyte adhesion. Taking into account the observations mentioned above, potential aggravating effects of these particles in a scenario of ongoing inflammation, for instance present in cardiovascular patients, need to be addressed further. Of note, the identified interaction between polystyrene microplastic particles and neutrophils in the peripheral blood is indicative of an innate immune response, the consequences of which require further investigations.

In summary, for the first time we here show that PS particles are capable of activating the endothelium with subsequent monocyte adhesion. Taking into consideration that this process is not only important for the immune response, but also a hallmark of the initiation of atherosclerosis, microplastics need be evaluated critically as a novel environmental risk factor for cardiovascular disease and a general risk assessment is needed.

## Supporting information

**S1 Fig. Raman spectrum and size measurement of PS particles.** (**A**) Raman spectrum of PS particles used in this study (**B**) Raman spectrum of PS particles as a reference, obtained at ramanlife.com. (**C**) Size distribution of PS particles were assessed by dynamic light scattering and presented as percent intensity.
(TIF)

**S2 Fig. Time- and dose-dependent cytokine expression of J774A.1 and MyEND cells after PS particle stimulation.** (**A**) Pilot study experiments (n = 2) in J774A.1 (top) and MyEND

(bottom) cells using different time points (3 and 6 hours with a concentration of $10^7$ particles/mL) and concentrations of PS particles ($10^3$, $10^5$ and $10^7$ particles/mL for 3 hours in J774.A1 cells and 6 hours in MyEND cells) to investigate kinetic and dosage effects. (**B**) Representative brightfield microscopy image showing a MyEND cell monolayer stimulated with PS particles at a concentration of $10^7$ particles/mL. Scale bar = 20 μm.
(TIF)

**S1 Data.**
(XLSX)

## Acknowledgments

We thank Daniela Beppler, Silke Brauschke, Michael Malysa and Sonja Hühn for excellent technical assistance and Dr. Katrin Roth, Imaging Core Facility Philipps University Marburg for excellent support for experiments under flow.

## Author Contributions

**Conceptualization:** Ann-Kathrin Vlacil, Bernhard Schieffer, Karsten Grote.

**Data curation:** Ann-Kathrin Vlacil, Nicole Trippel, Istemi Kuzu, Karsten Grote.

**Formal analysis:** Ann-Kathrin Vlacil, Sebastian Bänfer, Ralf Jacob, Karsten Grote.

**Funding acquisition:** Bernhard Schieffer, Karsten Grote.

**Investigation:** Ann-Kathrin Vlacil, Sebastian Bänfer, Karsten Grote.

**Methodology:** Ann-Kathrin Vlacil, Sebastian Bänfer, Ralf Jacob, Istemi Kuzu.

**Project administration:** Karsten Grote.

**Supervision:** Karsten Grote.

**Writing – original draft:** Ann-Kathrin Vlacil.

**Writing – review & editing:** Bernhard Schieffer, Karsten Grote.

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
