## [Decision Letter · Decision Letter 0]

31 Aug 2021

PONE-D-21-23667

Polystyrene microplastic particles induce endothelial activation

PLOS ONE

Dear Dr. Grote,

Thank you for submitting your manuscript to PLOS ONE. After careful consideration, we feel that it has merit but does not fully meet PLOS ONE’s publication criteria as it currently stands. Therefore, we invite you to submit a revised version of the manuscript that addresses the points raised during the review process.

We look forward to receiving your revised manuscript.

Kind regards,

Yi Cao

Academic Editor

PLOS ONE

Journal Requirements:

Reviewers' comments:

Reviewer's Responses to Questions

**Comments to the Author**

1. Is the manuscript technically sound, and do the data support the conclusions?

Reviewer #1: Partly

Reviewer #2: Yes

Reviewer #3: Yes

2. Has the statistical analysis been performed appropriately and rigorously? 

Reviewer #1: Yes

Reviewer #2: Yes

Reviewer #3: Yes

3. Have the authors made all data underlying the findings in their manuscript fully available?

Reviewer #1: Yes

Reviewer #2: Yes

Reviewer #3: Yes

4. Is the manuscript presented in an intelligible fashion and written in standard English?

Reviewer #1: No

Reviewer #2: Yes

Reviewer #3: Yes

5. Review Comments to the Author

Reviewer #1: In this study the authors demonstrate that polystyrene microplastics induce endothelial activation through in vivo and in vitro mouse model experiments. Although the results are very interesting, the study is very poorly presented with no real discussion of the data. Therefore, I request major revision for this study.

Following several requests:

The abstract is really badly written and poorly organized. For example:

Line 30: In my opinion, it is improper to use “substance” referring to plastic items. Please, revise the sentence.

Line 31: “Small particles”. What do you mean? Please, insert the size.

Line 31-36: Rather than talking about the microplastics classification (put it in the intro), it would be more appropriate to explain the premise of this work in detail. In addition, it would be important for readers to adds details on the methods (Have you performed an in vitro study? In vivo study? Etc.)

Line 50 and 55: Please, refer to the document “Plastic Europe, 2020” not to the website.

Line 54-55: At least one citation is required!

Line 55: Please, check the English sentence!

Line 57: Maybe the citation [3] is inappropriate. Please, there are thousands of citation that you can use.

Line 65-67: …PS also led to the aggregation and endothelial adhesion of RBC (Please, see Barshtein et al., 2016 doi: 10.1007/s12013-015-0705-6).

Line 80-81: You have used in your study carboxylated MPs but you never mentioned before. Please, check: Murano et al., 2021 (doi: 10.3389/fmars.2021.647394). Use it for explaining the relevance of carboxy-modified particles in environment.

Line 86: Why was such a high concentration of particles used in this study? Please, add an explanation.

Line 211: Just curiosity. Did you find the particles only in the liver or in other organs as well?

The discussions need to be expanded. There is no part of comparison with other studies in the literature or general considerations also made on marine organisms for example.

Another suggestion is related to the title: why use the word “particles” if microplastic is a particle?

Reviewer #2: Dear Authors,

this is an interesting manuscript which for the first time highlights, through in vitro experiments and using mouse models, some new aspects of the deleterious effects of microplastics at the vascular level. In particular, the Authors found that polystyrene microplastic particles (PS) induce activation of endothelial cells (increased expression of adhesion molecules and leukocyte interaction) and increases the expression and release of inflammatory cytokines from monocytic cells. They also demonstrate PS uptake by circulating neutrophils.

Although the topic is very interesting, some points are not clear.

Here are my comments:

• MATERIALS AND METHODS: Line 73: The "MyEND" cells as reported in reference number 15 is a cell line: it should be specified. Line 81: TRICT is an acronym and I suggest specifying Tetramethylrhodamine the first time it is mentioned. Line 96: An important point that was not mentioned and that the Authors must specify is how many animals were used, this is important for the data statistical significance, especially for the experiments performed on blood, liver and aortic tissue samples. Line 109: which genes were evaluated, and which primers were used? It must be specified in the methods before finding them cited in the results and in the figures legends.

• RESULTS: Figures S2: The Authors performed only 2 experiments even though 3 experiments would have been better to reach statistical significance but defining it as a pilot study this could be accepted. However, in the legend it is not clear at 3 and 6 hours which PS concentration was used, specify as in the results 107. In J774A.1 cells, interleukin 1beta increases more at 3 hours than at 6 hours which is like the control; how do the authors justify this? Why did the authors choose 3 and 6 hours? It would have been interesting to evaluate longer stimulation times to visualize a chronic effect (like 24, 48, 72 hours of stimulation) which is what happens in real life. In fact, the chronicity of encountering microplastics leads to diseases. The authors have discussed this (Lines 263-266) but they could expand this concept by adding some references about it as well.

Figure 1: Does n = 7-10 refer to the number of experiments conducted?

Figure 2: Authors should show more representative images of the reported data (especially for B)

Figure 3A: It is low resolution, improve the resolution. Line 213 add a reference at the end of the sentence "indicating a hepatic inflammatory response to the PS particles".

Figure 4 and Line 229: ICAM-1 does not reach statistical significance, please specify that it is a trend.

Line 238: does n = 7 / group mean 7 mice per group? see comment on the number of mice in the comments on materials and methods.

Reviewer #3: In this study Ann-Kathrin Vlacil and collaborators demonstrate that microplastics are new environmental risk factors for endothelial inflammation.

The topic of this paper is very interesting and data convincing. Only few issues should be addressed in order to improve this study for the readers of PlosOne:

1. In Material and Methods, “Cell Culture and Treatment” paragraph, cells were exposed for 3 and 6 hours to PS particles (as also reported in suppl figure 2), instead in “Cell viability assay” and “adhesion assay” the exposure is for 16 hours. Please better clarify the exposure time to PS particles. What is the rationale in choosing 3, 6 and 16 hours? Probably a time course with the addition of longer exposure could be appropriate. Please provide data in case author have performed experiments of longer exposure.

2. In figure 2 images of the total well representative of a bigger field should be added to have a more consistent idea of cell adhesion.

6. PLOS authors have the option to publish the peer review history of their article (what does this mean?). If published, this will include your full peer review and any attached files.

Reviewer #1: No

Reviewer #2: No

Reviewer #3: No

---

## [Author Response · Author response to Decision Letter 0]

19 Oct 2021

We thank all reviewers for their valuable suggestions and corrections that helped us to improve the quality of our manuscript. All concerns/comments have been addressed properly. Please find below our detailed response and the corrections that have been made. All corrections are highlighted in red color in the revised version of our manuscript. Thank you very much for consideration.

Reviewer #1

Reviewer: The abstract is really badly written and poorly organized. For example:

Line 30: In my opinion, it is improper to use “substance” referring to plastic items. Please, revise the sentence.

Authors: We have replaced the term substance with resource (line 30) and have better organized the abstract according to the reviewer's suggestions.

Reviewer: Line 31: “Small particles”. What do you mean? Please, insert the size.

Authors: We now provide a general definition of microplastics (line 31-33) and defined the particles used in this study more precisely (1 µm, line 36). In addition, we have performed size measurements of the used particles by dynamic light scattering to confirm the manufacturer's information (S1C Fig, line 90-92, line 400-401).

Reviewer: Line 31-36: Rather than talking about the microplastics classification (put it in the intro), it would be more appropriate to explain the premise of this work in detail. In addition, it would be important for readers to adds details on the methods (Have you performed an in vitro study? In vivo study? Etc.)

Authors: As suggested, we have moved the classification of microplastics, etc. to the introduction, described the intent of our study (line 35-38), and described the methods used in more detail (line 37-38).

Reviewer: Line 50 and 55: Please, refer to the document “Plastic Europe, 2020” not to the website.

Authors: We now refer to the document (line 54/55 and 59, ref#2).

Reviewer: Line 54-55: At least one citation is required!

Authors: We have restructured this part of the introduction. The information on the size of the particles is covered by new ref#1 (line 48-52).

Reviewer: Line 55: Please, check the English sentence!

Authors: We have rewritten this sentence (line 59-60).

Reviewer: Line 57: Maybe the citation [3] is inappropriate. Please, there are thousands of citation that you can use.

Authors: The paper was replaced by Isaac MN et al. (line 61, new ref#5).

Reviewer: Line 65-67: …PS also led to the aggregation and endothelial adhesion of RBC (Please, see Barshtein et al., 2016 doi: 10.1007/s12013-015-0705-6).

Authors: Thank you for pointing this out, we have included the findings and the reference (line 70-72, ref#14).

Reviewer: Line 80-81: You have used in your study carboxylated MPs but you never mentioned before. Please, check: Murano et al., 2021 (doi: 10.3389/fmars.2021.647394). Use it for explaining the relevance of carboxy-modified particles in environment.

Authors: We agree with the reviewer that this is an important information and have now regularly indicated in the manuscript that we used carboxylated PS particles (e.g. line 36, 74 etc.). We have also cited the suggested literature (ref#21) and a review article on that topic by Andrady AL (ref#22) and discussed the relevance of surface modification and charge (line 274-278).

Reviewer: Line 86: Why was such a high concentration of particles used in this study? Please, add an explanation.

Authors: We decided for this concentration according to our pilot dose dependency and time response study in vitro (Suppl. Fig. 2), showing greatest effects at 107/mL. This concentration corresponds to 5.4 �g/mL. Respective information for all used PS particle concentrations (pilot studies, cell viability) were now added to the Materials and Methods section (line 94-95) to facilitate comparability with other studies. In view of this, we used a rather lower particle concentration in our study (see e.g. ref#17 Wu et al. 2020 using 12.5 and 50 mg/mL). As can be seen in S2BFig, each cell is in contact with only a few particles at this concentration, which we have already described in the results section (line 196-199).

Reviewer: Line 211: Just curiosity. Did you find the particles only in the liver or in other organs as well?

Authors: So far, we have only examined a handful of organs/tissues in this regard. We found particles in spleen, bone marrow and adipose tissue as well. A comprehensive analysis of further organs/tissues is planned for the future. We provide some exemplarily pictures for the reviewer only in the attached word file.

Reviewer: The discussions need to be expanded. There is no part of comparison with other studies in the literature or general considerations also made on marine organisms for example.

Authors: We now provide a more detailed comparison with other studies in the literature, see line 269-274 (ref#19 and ref#20), line 274-275(ref#8 and ref#21), line 299-302 (ref#17 and ref#12) or line 305-310 (ref#8, ref#23, ref#24 and ref#25).

Reviewer: Another suggestion is related to the title: why use the word “particles” if microplastic is a particle?

Authors: The term microplastic particles is frequently used in the field (more than 2000 hits in PubMed). Furthermore, we believe that this name best reflects the type of our stimulation: namely, with a variety of defined particles.

Reviewer #2

Reviewer: MATERIALS AND METHODS: Line 73: The "MyEND" cells as reported in reference number 15 is a cell line: it should be specified. 

Authors: Thank you for pointing this out, we now have clearly indicated in material and methods that MyEND cells as well as J77A4.1 are cell lines. In addition, we provided the accession where more details about the cells can be found (line 79-81).

Reviewer: Line 81: TRICT is an acronym and I suggest specifying Tetramethylrhodamine the first time it is mentioned. 

Authors: We have specified TRITC (line 87-88).

Reviewer: Line 96: An important point that was not mentioned and that the Authors must specify is how many animals were used, this is important for the data statistical significance, especially for the experiments performed on blood, liver and aortic tissue samples.

Authors: This is certainly an important information. Since we show the individual data points in the figures, we had omitted this information in the first version of the manuscript. However, it is often difficult to identify the exact number from the figures. We have now indicated the animal numbers used (line 107). Please note, the number of animals may vary between figures because we did not use all mice in all experiments or if we had to exclude data based on CT values in the PCR analyses, or similar.

Reviewer: Line 109: which genes were evaluated, and which primers were used? It must be specified in the methods before finding them cited in the results and in the figures legends.

Authors: We now specified the genes which were evaluated by real-time PCR and listed the sequences of the corresponding primers in materials and methods (line 125-136).

Reviewer: RESULTS: Figures S2: The Authors performed only 2 experiments even though 3 experiments would have been better to reach statistical significance but defining it as a pilot study this could be accepted. However, in the legend it is not clear at 3 and 6 hours which PS concentration was used, specify as in the results 107. In J774A.1 cells, interleukin 1beta increases more at 3 hours than at 6 hours which is like the control; how do the authors justify this? Why did the authors choose 3 and 6 hours? It would have been interesting to evaluate longer stimulation times to visualize a chronic effect (like 24, 48, 72 hours of stimulation) which is what happens in real life. In fact, the chronicity of encountering microplastics leads to diseases. The authors have discussed this (Lines 263-266) but they could expand this concept by adding some references about it as well.

Authors: To be more precise, we now specify the stimulation concentration (107 particles/mL) and time points (3 and 6 hours), see line 405-407 in figure legend and line 195-196 in the result section. In these pilot experiments, we examined two early time points and selected the time point where we saw the best effects (3 hours for J774A.1 and 6 hours for MyEND). We have focused on the initial onset of pro-inflammatory cytokine expression as a clear result of microplastic stimulation and wanted to exclude secondary effects potentially arising during long time exposure. Long time experiments with both cell types under our culture conditions are not feasible because cells begin to detach after more than 24 hours (independent of PS particles). However, we of course agree with the reviewer that organisms are exposed to microplastic in a chronic manner and added two recent studies investigating long-term exposure in a marine ecotoxicological model organism (see line 269-274 and ref#19 and ref#20).

Reviewer: Figure 1: Does n = 7-10 refer to the number of experiments conducted?

Authors: Yes, the number of independent experiments is given.

Reviewer: Figure 2: Authors should show more representative images of the reported data (especially for B)

Authors: We have chosen larger image details or selected other images (please see also reviewer #3). The adherent monocytes on the images now better reflect the differences found in the quantitate evaluation, namely that PS increases the adherence of monocytes about 3-fold.

Reviewer: Figure 3A: It is low resolution, improve the resolution. 

Authors: We have increased both the resolution and the image size to better identify the PS particles in the tissue.

Reviewer: Line 213 add a reference at the end of the sentence "indicating a hepatic inflammatory response to the PS particles".

Authors: We already had a reference to the hepatic inflammatory response (Sack et al., ref#16), which we have now given here. In addition, we extended the discussion in this context (line 302-305).

Reviewer: Figure 4 and Line 229: ICAM-1 does not reach statistical significance, please specify that it is a trend.

Authors: To be more precise, we now state that Icam-1 shows a trend towards elevated expression (line 251-252).

Reviewer: Line 238: does n = 7 / group mean 7 mice per group? See comment on the number of mice in the comments on materials and methods.

Authors: Yes, the indicated numbers refer to the number of mice per group. We have addressed this issue in material and methods as suggested, see line 107 and in addition we now specify this in more detail in the figure legends (line 243 and 260).

Reviewer #3

Reviewer: In Material and Methods, “Cell Culture and Treatment” paragraph, cells were exposed for 3 and 6 hours to PS particles (as also reported in suppl figure 2), instead in “Cell viability assay” and “adhesion assay” the exposure is for 16 hours. Please better clarify the exposure time to PS particles. What is the rationale in choosing 3, 6 and 16 hours? Probably a time course with the addition of longer exposure could be appropriate. Please provide data in case author have performed experiments of longer exposure.

Authors: As stated above, we choose the indicated stimulation time points according to our pilot study, showing highest mRNA expression of inflammatory cytokines after 3 hours (J774A.1) and 6 hours (MyEND). We discuss this in the results section (see line 193-196). In the adhesion assay, we chose 16 hours to allow adequate expression of the adhesion molecules at the protein level (translation and transport to the membrane). In addition, we have already had good experience with this timing in previous studies (e.g. doi: 10.3390/cells10082146). For the cell toxicity study, we again chose the same time period to exclude cytotoxic effects of the PS particles on the endothelial cells within these 16 hours of stimulation for the adhesion assay. We have now indicated this in the results section (see line 192-193 and line 216-218). In this study, we focused exclusively on effects of PS particles at early time points and not on longer effects in which secondary effects of PS particles may also play a role. In addition, long time experiments with both cell types under our culture conditions are not feasible because – independent of PS particles – cells start to detach after more than 24 hours. However, longer exposure times in the environment are obviously relevant, so we now discuss two already existing studies on long-term effects of microplastics on marine organisms (see line 269-274 and ref#19 and ref#20 and see also reviewer #2).

Reviewer: In figure 2 images of the total well representative of a bigger field should be added to have a more consistent idea of cell adhesion.

Authors: We have enlarged the image sections shown in both cases. In B (flow experiments), however, only slightly, because we were limited by the equipment of the flow chamber and microscope in terms of resolution and detection of the cells. The images were taken at the end of the washing step under flow conditions. We now show the actual image size on which the evaluations were made (please see also reviewer #2).

---

## [Decision Letter · Decision Letter 1]

1 Nov 2021

PONE-D-21-23667R1Polystyrene microplastic particles induce endothelial activationPLOS ONE

Dear Dr. Grote,

Thank you for submitting your manuscript to PLOS ONE. After careful consideration, we feel that it has merit but does not fully meet PLOS ONE’s publication criteria as it currently stands. Therefore, we invite you to submit a revised version of the manuscript that addresses the points raised during the review process.

ACADEMIC EDITOR: Please make a minor revision according to the comments of reviewer 1 before the final acceptance of your manuscript. 

We look forward to receiving your revised manuscript.

Kind regards,

Yi Cao

Academic Editor

PLOS ONE

Journal Requirements:

Reviewers' comments:

Reviewer's Responses to Questions

**Comments to the Author**

1. If the authors have adequately addressed your comments raised in a previous round of review and you feel that this manuscript is now acceptable for publication, you may indicate that here to bypass the “Comments to the Author” section, enter your conflict of interest statement in the “Confidential to Editor” section, and submit your "Accept" recommendation.

Reviewer #1: All comments have been addressed

Reviewer #2: All comments have been addressed

Reviewer #3: All comments have been addressed

2. Is the manuscript technically sound, and do the data support the conclusions?

Reviewer #1: Yes

Reviewer #2: Yes

Reviewer #3: Yes

3. Has the statistical analysis been performed appropriately and rigorously? 

Reviewer #1: Yes

Reviewer #2: Yes

Reviewer #3: Yes

4. Have the authors made all data underlying the findings in their manuscript fully available?

Reviewer #1: Yes

Reviewer #2: Yes

Reviewer #3: Yes

5. Is the manuscript presented in an intelligible fashion and written in standard English?

Reviewer #1: Yes

Reviewer #2: Yes

Reviewer #3: Yes

6. Review Comments to the Author

Reviewer #1: The authors have adequately addressed my previous comments and improved the quality of the manuscript. However, some small issues could be addressed.

-First of all, in my opinion, even define plastic as a "resource" is somehow incorrect. Therefore, the authors have to review the first sentence of the abstract. Plastic is a material/a product ....

-"In vivo" and "in vitro" must be must be written to Italic character

-Check References

I really appreciate the example images And I hope the authors will provide further insights on this topic for future publications.

Reviewer #2: The Authors have been addressed all comments. Hence, the manuscript has been improved and I have not further comment.

Reviewer #3: (No Response)

7. PLOS authors have the option to publish the peer review history of their article (what does this mean?). If published, this will include your full peer review and any attached files.

Reviewer #1: No

Reviewer #2: No

Reviewer #3: No

---

## [Author Response · Author response to Decision Letter 1]

2 Nov 2021

Reviewer #1

We thank Reviewer#1 to further improve the quality of our manuscript. All corrections are highlighted in red color in the revised version of our manuscript.

Reviewer: First of all, in my opinion, even define plastic as a "resource" is somehow incorrect. Therefore, the authors have to review the first sentence of the abstract. Plastic is a material/a product....

Authors: That is true, mineral oil would be the raw material in this context. We have decided to use the term material (line 30).

Reviewer: "In vivo" and "in vitro" must be written to Italic character.

Authors: We have now italicized both terms throughout the manuscript.

Reviewer: Check References.

Authors: We have checked again the correct use of all references.

---

## [Editor Report · Decision Letter 2]

4 Nov 2021

Polystyrene microplastic particles induce endothelial activation

PONE-D-21-23667R2

Dear Dr. Grote,

We’re pleased to inform you that your manuscript has been judged scientifically suitable for publication and will be formally accepted for publication once it meets all outstanding technical requirements.

Kind regards,

Yi Cao

Academic Editor

PLOS ONE
---

## [Editor Report · Acceptance letter]

8 Nov 2021

PONE-D-21-23667R2 

Polystyrene microplastic particles induce endothelial activation 

Dear Dr. Grote:

I'm pleased to inform you that your manuscript has been deemed suitable for publication in PLOS ONE. Congratulations! Your manuscript is now with our production department. 

Kind regards, 

on behalf of

Dr. Yi Cao 

Academic Editor

PLOS ONE